# Neural Additive Models:
# Interpretable Machine Learning with Neural Nets

**Rishabh Agarwal**[*]
Google Research, Brain Team

**Levi Melnick**
Microsoft Research

**Nicholas Frosst**
Cohere

**Xuezhou Zhang**
University of Wisconsin-Madison

**Ben Lengerich**
MIT

**Rich Caruana**
Microsoft Research

**Geoffrey E. Hinton**
Google Research, Brain Team

## Abstract

Deep neural networks (DNNs) are powerful black-box predictors that have achieved impressive performance on a wide variety of tasks. However, their accuracy comes at the cost of intelligibility: it is usually unclear how they make their decisions. This hinders their applicability to high stakes decision-making domains such as healthcare. We propose Neural Additive Models (NAMs) which combine some of the expressivity of DNNs with the inherent intelligibility of generalized additive models. NAMs learn a linear combination of neural networks that each attend to a single input feature. These networks are trained jointly and can learn arbitrarily complex relationships between their input feature and the output. Our experiments on regression and classification datasets show that NAMs are more accurate than widely used intelligible models such as logistic regression and shallow decision trees. They perform similarly to existing state-of-the-art generalized additive models in accuracy, but are more flexible because they are based on neural nets instead of boosted trees. To demonstrate this, we show how NAMs can be used for multitask learning on synthetic data and on the COMPAS recidivism data due to their composability, and demonstrate that the differentiability of NAMs allows them to train more complex interpretable models for COVID-19. Source code is available at neural-additive-models.github.io.

## 1 Introduction

While deep neural networks have achieved impressive results on tasks such as computer vision [17] and language modeling [31], it is notoriously difficult to understand how such networks make predictions, and they are often considered as black-box models. This hinders their applicability to high-stakes domains such as healthcare, finance and criminal justice. Various efforts have been made to demystify the predictions of neural networks (NNs). For example, one family of methods, represented by LIME [33], attempt to *explain* individual predictions of a neural network by approximating it locally with interpretable models such as linear models and shallow trees[2]. However, these approaches often fail to provide a global view of the model and their explanations often are not faithful to what the original model computes or do not provide enough detail to understand the model's behavior [35].

---

[*]Correspondence to: Rishabh Agarwal <rishabhagarwal@google.com>, Levi Melnick <lemeln@microsoft.com>, and Rich Caruana <rcaruana@microsoft.com>.

[2]Linear models, shallow decision trees and GAMs are interpretable only if the features they are trained on are interpretable.

35th Conference on Neural Information Processing Systems (NeurIPS 2021).

In this paper, we make restrictions on the *structure* of neural networks, which yields a family of glass-box models called Neural Additive Models (NAMs), that are inherently interpretable while suffering little loss in prediction accuracy when applied to tabular data. Methodologically, NAMs belong to a model family called Generalized Additive Models (GAMs) [14]. GAMs have the form:

$$g(\mathbb{E}[y]) = \beta + f_1(x_1) + f_2(x_2) + \cdots + f_K(x_K) \tag{1}$$

where $\mathbf{x} = (x_1, x_2, \ldots, x_K)$ is the input with $K$ features, $y$ is the target variable, $g(.)$ is the link function (*e.g.,* logistic function) and each $f_i$ is a univariate shape function with $\mathbb{E}[f_i] = 0$. Generalized linear models, such as logistic regression, are a special form of GAMs where each $f_i$ is restricted to be linear.

NAMs learn a linear combination of networks that each attend to a single input feature: each $f_i$ in (1) is parametrized by a neural network. These networks are trained jointly using backpropagation and can learn arbitrarily complex shape functions. Interpreting NAMs is easy as the impact of a feature on the prediction does not rely on the other features and can be understood by visualizing its corresponding shape function (*e.g.,* plotting $f_i(x_i)$ *vs.* $x_i$). While interpretability of NAMs may seem heuristic, the graphs learned by NAMs are an exact description of how NAMs compute a prediction.

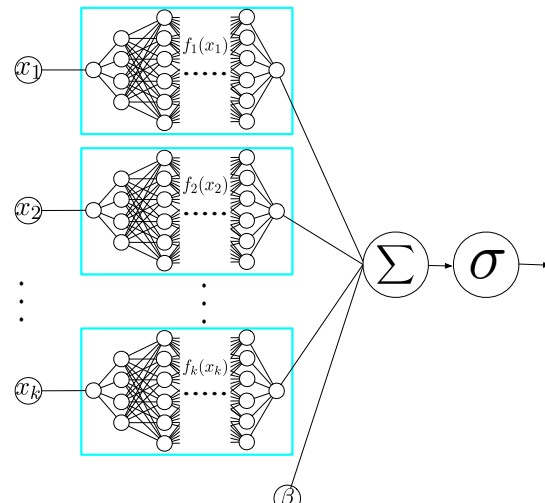

Figure 1: NAM architecture for binary classification. Each input variable is handled by a different neural network. This results in easily interpretable yet highly accurate models.

Traditionally, GAMs were fitted via iterative backfitting using smooth low-order splines, which reduce overfitting and can be fit analytically. More recently, GAMs [5] were fitted with boosted decision trees to improve accuracy and to allow GAMs to learn jumps in the feature shaping functions to better match patterns seen in real data that smooth splines could not easily capture. This paper examines using DNNs to fit generalized additive models (NAMs) which provides the following advantages:

- NAMs introduce an expressive yet intelligible class of models to the deep learning (DL) community, a much larger community than the one using tree-based GAMs.

- NAMs are likely to be combined with other DL methods in ways we don't foresee. This is important because a key drawback of deep learning is interpretability. For example, NAMs have already been employed for survival analysis [46].

- NAMs, due to the flexibility of NNs, can be easily extended to various settings problematic for boosted decision trees. For example, extending boosted tree GAMs to multitask, multi-class or multi-label learning requires significant changes to how trees are trained, but is easily accomplished with NAMs without requiring changes to how neural nets are trained due to their composability (Section 4.2). Futhermore, the differentiability of NAMs allows them to train more complex interpretable models for COVID-19 (Section 4.1).

- Graphs learned by NAMs are not just an explanation but an exact description of how NAMs compute a prediction. As such, a decision-maker can easily interpret NAMs and understand exactly how they make decisions. This would help harness the expressivity of neural nets on high-stakes domains with intelligibility requirements, *e.g.,* in-hospital mortality prediction [22].

- NAMs are more scalable as inference and training can be done on GPUs/TPUs or other specialized hardware using the same toolkits developed for deep learning over the past decade – GAMs currently cannot.

- Accurate GAMs [5] currently require millions of decision trees to fit each shape function while NAMs only use a small ensemble (2 - 100) of neural nets. Thus, NAMs are relatively much easier to extend compared to GAMs.

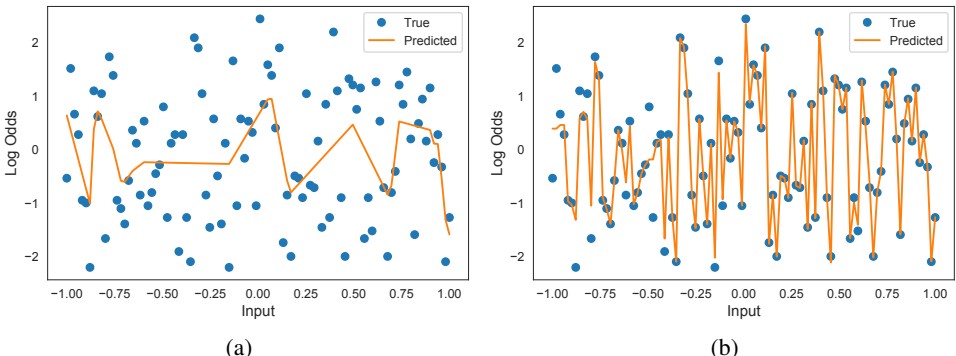

Figure 2: **Accurately Fitting the Toy Dataset**: Training predictions learned by a single hidden layer neural network with 1024 (a) standard ReLU, and (b) ReLU-$n$ with ExU hidden units trained for 10,000 epochs on the binary classification dataset described in Section 2. We can see that the ReLU network has learned a fairly smooth function while the ExU network has learned a very jumpy function. We find that a DNN with three hidden layers also learned smooth functions (see Figure A.3).

## 2 Neural Additive Models

**Modeling jagged shape functions** is required to learn accurate additive models as there are often sharp jumps in real-world datasets, *e.g.,* see Figure 4 for jumps in graphs for PFRatio and Bilirubin which correspond to real patterns in the MIMIC-II dataset [38] (Section A.1). Similarly, Caruana et al. [5] observe that GAMs fit using splines tend to over regularize and miss genuine details in real data, yielding less accuracy than tree-based GAMs. Therefore, we require that neural networks (NNs) are able to learn highly non-linear shape functions, to fit these patterns.

Although NNs can approximate arbitrarily complex functions [18], we find that standard NNs fail to model highly jumpy 1D functions, and demonstrate this failure empirically using **a toy dataset**. The toy dataset is constructed as follows: For the input $x$, we sample 100 evenly spaced points in [-1, 1]. For each $x$, we sample $p$ uniformly random in [0.1, 0.9) and generate 100 labels from a Bernoulli random variable which takes the value 1 with probability $p$. This creates a binary classification dataset of $(x, y)$ tuples with 10,000 points. Figure 2 shows the log-odds of the empirical probability $p$ (*i.e.,* $\log \frac{p}{1-p}$) of classifying the label of $x$ as 1 for each input $x$. This dataset tests the NN's ability to "overfit" the data, rather than its ability to generalize.

Over-parameterized NNs with ReLUs [25] and standard initializations such as Kaiming initialization [16] and Xavier initialization [10] struggle to overfit this dataset when trained using mini-batch gradient descent, despite the NN architecture being expressive enough[3](see Figures 2(a) and A.3). This difficulty of learning large local fluctuations with ReLU networks without affecting their global behavior when fitting jagged functions might be due to their bias towards smoothness [2, 32].

We propose *exp-centered* (ExU) hidden units to overcome this neural net failure: we simply learn the weights in the logarithmic space with inputs shifted by a bias. Specifically, for a scalar input $x$, each hidden unit using an activation function $f$ computes $h(x)$ given by

$$h(x) = f\left(e^w * (x - b)\right) \qquad (2)$$

where $w$ and $b$ are the weight and bias parameters. The intuition behind ExU units is as follows: For modeling jagged functions, a hidden unit should be able to change its output significantly, with a tiny change in input. This requires the unit to have extremely large weight values depending on the sharpness of the jump. The ExU unit computes a linear function of input where the slope can be very steep with small weights, making it easier to modify the output easily during training. ExU units do not improve the expressivity of neural nets, however they do improve their learnability for fitting jumpy functions. While we use ExU units to train accurate NAMs, they are more generally applicable for approximating jumpy functions with neural nets.

We noticed that ExU units with standard weight initialization also struggle to learn jagged curves; instead initializing the weights using a normal distribution $\mathcal{N}(x, 0.5)$ with $x \in [3, 4]$ works well in practice. This initialization simply ensures that the initial network starts with a jagged (random)

---

[3]This problem doesn't occur with full-batch gradient descent.

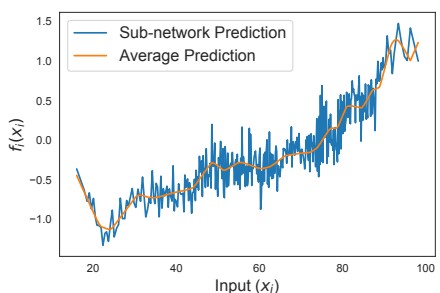

Figure 3: **Regularizing ExU networks.** Output of a ExU feature net trained with dropout = 0.2 for the age feature in the MIMIC-II dataset [38]. Predictions from individual subnets (as a result of dropping out hidden units) are much more jagged than the average predictions using the entire feature net. Refer to Section A.3 for an overview of regularization approaches used in this work.

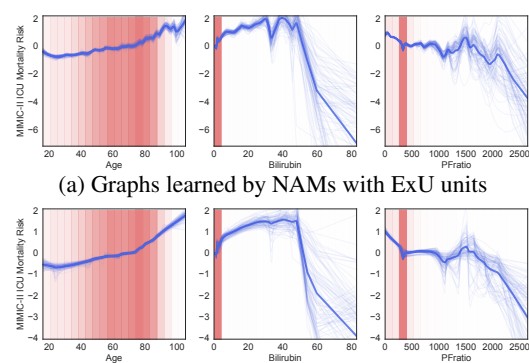

(a) Graphs learned by NAMs with ExU units

(b) Graphs learned by NAMs with standard units

Figure 4: **ExU** *vs.* **standard** hidden units. On MIMIC-II, NAMs trained with ExU units learn jumpier graphs than with standard units while achieving a similar AUC ($\approx 0.829$). Ensembling them further improves performance ($\approx 0.830$). Note that white regions in the plots correspond to regions with low data density (typically a few points) and thus we see much higher variance in the learned shape functions. We present a detailed case study on the MIMIC-II dataset in Section A.1.

function which we empirically find to be crucial for fitting any jumpy function. Furthermore, we use ReLU activations capped at $n$ (ReLU-$n$) [21] to ensure that each ExU unit is active in a small input range, making it easier to model sharp jumps in a function without significantly affecting the global behavior. ExU-units can be combined with any activation function (*i.e.*, any $f$ can be used in (2)), but ReLU-$n$ performs well in practice. Figure 2(b) shows that NNs with ExU units are able to fit the toy dataset significantly better than standard NNs.

Finally, realistic shape functions typically tend to be smooth with large jumps at only a few points (Figure 4). To avoid overfitting with ExUs, strong regularization is crucial which can learn such realistic functions (*e.g.,* Figure 3). With ReLUs, we can typically fit smooth functions but they might miss some of these jumps. To avoid overfitting when fitting NAMs with ExUs, we employ various regularization methods including dropout, weight decay, output penalty, and feature dropout (see Section A.3 for an overview).

## 2.1 Intelligibility and Modularity of NAMs

The intelligibility of NAMs results in part from the ease with which they can be visualized. Because each feature is handled independently by a learned shape function parameterized by a neural net, one can get a full view of the model by simply graphing the individual shape functions. For data with a small number of inputs, it is possible to have an accessible explanation of the model's behavior visualized fully on a single page. Please note these shape function plots are not just an explanation but an *exact* description of how NAMs compute a prediction. A decision-maker can easily interpret such models and understand exactly how they make decisions, for example, we validated the behavior of NAMs on the MIMIC-II dataset [38] with a doctor (Appendix A.1).

We set the average score for each graph (*i.e.,* each feature) averaged across the entire training dataset to zero by subtracting the mean score. To make individual shape functions identifiable and modular, a single bias term is then added to the model so that the average predictions across all data points matches the observed baseline. This makes interpreting the contribution of each term easier: *e.g.,* on binary classification tasks, negative scores decrease probability, and positive scores increase probability compared to the baseline probability of observing that class. This property also allows each graph to be removed from the NAM (zeroed out) without introducing bias to the predictions.

**Visualization**. We plot each shape function and the corresponding data density on the same graph. Specifically, we plot each learned shape function $f_k(x_k)$ *vs.* $x_k$ for an ensemble of NAMs using a semi transparent blue line, which allows us to see when the models in the ensemble learned the same shape function and when they diverged. This provides a sense of the confidence of the learned shape functions. We also plot on the same graphs the normalized data density, in the form of pink bars. The darker the shade of pink, the more data there is in that region. This allows us to know when the model had adequate training data to learn appropriate shape functions.

Table 1: **Single-task learning NAM results**. Means and standard deviations are reported from 5-fold cross validation. Higher AUCs and lower RMSEs are better. We report results on two widely used regression datasets, namely California Housing [27] for predicting housing prices and FICO [9] for understanding credit score predictions, as well as two classification datasets, namely Credit [7] for financial fraud detection and MIMIC-II [38] for predicting mortality in ICUs. We present a case study on the MIMIC-II dataset in Section A.1 and discuss the interpretations from NAMs on other datasets in Section A.2.

| Model | MIMIC-II (AUC) | Credit (AUC) | CA Housing (RMSE) | FICO (RMSE) |
|---|---|---|---|---|
| Log./Linear Reg. | $0.791 \pm 0.007$ | $0.975 \pm 0.010$ | $0.728 \pm 0.015$ | $4.344 \pm 0.056$ |
| CART | $0.768 \pm 0.008$ | $0.956 \pm 0.004$ | $0.720 \pm 0.006$ | $4.900 \pm 0.113$ |
| NAMs | $0.830 \pm 0.008$ | $0.980 \pm 0.002$ | $0.562 \pm 0.007$ | $3.490 \pm 0.081$ |
| EBMs | $0.835 \pm 0.007$ | $0.976 \pm 0.009$ | $0.557 \pm 0.009$ | $3.512 \pm 0.095$ |
| XGBoost | $0.844 \pm 0.006$ | $0.981 \pm 0.008$ | $0.532 \pm 0.014$ | $3.345 \pm 0.071$ |
| DNNs | $0.832 \pm 0.009$ | $0.978 \pm 0.003$ | $0.492 \pm 0.009$ | $3.324 \pm 0.092$ |

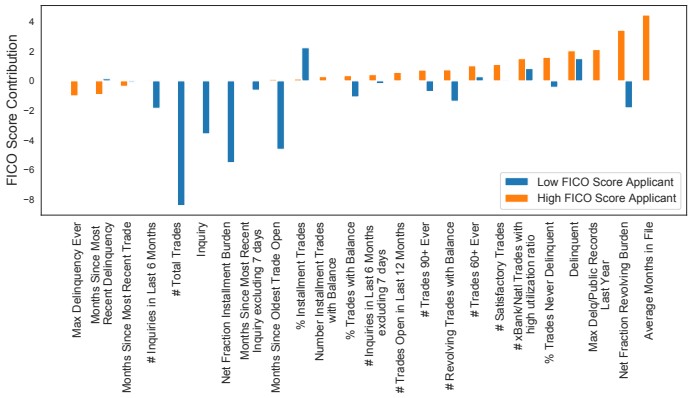

Figure 5: **Understanding individual predictions for credit scores**. Feature contribution using the learned NAMs for predicting scores of two applicants in the FICO dataset [9]. For a given input, each feature net in the NAM acts as a lookup table and returns a contribution term. These contributions are combined in a modular way: they are added up, and passed through a link function for prediction. the longer a person's credit history, the better it is for their credit score The high scoring applicant has a long credit history (Average Months on File), which contributes to their credit score better. On the contrary, the low scoring applicant used their credit quite frequently (Total Number of Trades) and has a large burden (Net Fraction Installment Burden), thus resulting in a low score.

Figure 6: **California Housing**. Graphs learned by NAMs trained to predict house prices [27] for two most important features. As expected, The house prices increase linearly with median income in high data density regions. Furthermore, the graph for longitude shows sharp jumps in price prediction around the location of San Francisco and Los Angeles.

## 3 Evaluating the Accuracy of NAMs

In this section, we evaluate the single-task learning capacity of NAMs against the following baselines on both regression and classification tasks:

- **Logistic / Linear Regression and Decision Trees (CART)**: Prevalent intelligible models. For both methods above we use the `sklearn` implementation [28], and tune the hyper-parameters with grid search.

- **Explainable Boosting Machines (EBMs)**: Current state-of-the-art GAMs [5, 23] which use gradient boosting of millions of shallow bagged trees that cycle one-at-a-time through the features.

- **Deep Neural Networks (DNNs)**: Unrestricted, full-complexity models which can model higher-order interaction between the input features. This gives us a sense of how much accuracy we sacrifice in order to gain interpretability with NAMs.

- **Gradient Boosted Trees (XGBoost)**: Another class of full-complexity models that provides an upper bound on the achievable test accuracy in our experiments. We use the XGBoost implementation [6].

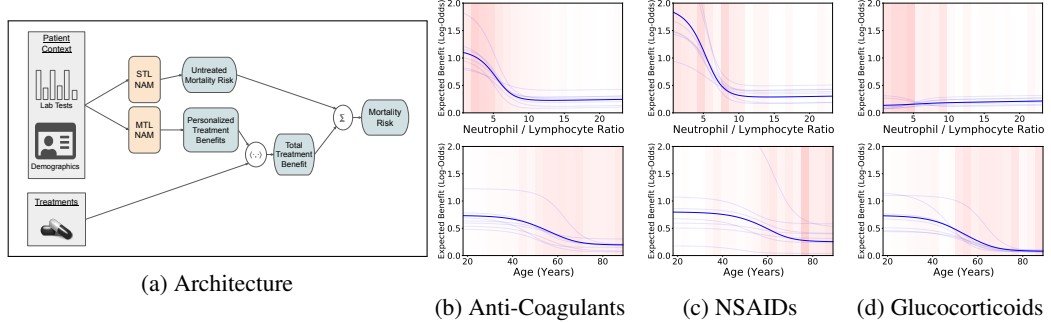

(a) Architecture

(b) Anti-Coagulants    (c) NSAIDs    (d) Glucocorticoids

Figure 7: **Estimating personalized treatment benefits for Covid-19 patients.** NAMs provide a unique combination of intelligibility and differentiability which make them suitable as a component in contextual parameter generation (a). By applying NAMs in this way, we are able to estimate and interpret personalized benefits of medical treatments for Covid-19 patients (b-d).

**Training and Evaluation**. Feature nets in NAMs are selected amongst (1) DNNs containing 3 hidden layers with 64, 64 and 32 units and ReLU activation, and (2) single hidden layer NNs with 1024 ExU units and ReLU-1 activation. We perform 5-fold cross validation to evaluate the accuracy of the learned models. To measure performance in Table 1, we use area under the precision-recall curve (AUC) for binary classification and root mean-squared error (RMSE) for regression. More details about training and evaluation protocols can be found in Section A.5 in the appendix.

NAMs achieve comparable performance to EBMs on both classification and regression datasets, making them a competitive alternative to EBMs. Given this observation, we next look at some additional capabilities of NAMs that are not available to EBMs or any tree-based learning methods.

## 4 Unique Capabilities of NAMs

### 4.1 Intelligible Parameter Generation: Leveraging the Differentiability of NAMs

Medical treatment protocols are designed to deliver treatments to patients who would most benefit from them. To optimize treatment protocols, we would like a model which provides an intelligible map from patient information to an estimate of benefit for each potential treatment. To accomplish this, we use a NAM to generate parameters for personalized models of mortality risk given treatment (Fig. 7). By training to match predicted mortality risk with observed mortality, the NAM encodes expected treatment benefits as a function of patient information. NAMs are the only nonlinear GAM suitable for this application because NAMs are differentiable and can be trained via backpropagation.

Figure 7 shows a NAM trained to predict treatment benefits for Covid-19 patients. We train the model on deidentified data from over 3000 Covid-19 patients. The model suggests that the benefits of anti-coagulants and NSAIDs decrease with increased Neutrophil / Lymphocyte Ratio (NLR), while the effectiveness of glucocorticoids slightly increases with increasing NLR. NLR is a marker of inflammation and severe Covid-19; it is thus expected that anti-coagulants (which target a distinct biomedical pathway) and NSAIDs (which are weaker) would not be as effective for patients with elevated NLR. In contrast, glucocorticoids become more effective for patients with more inflammation. This example shows the utility of a *differentiable* nonlinear additive model such as NAMs.

### 4.2 Multitask Learning

One advantage of NAMs is that they are easily extended to multitask learning (MTL) [4], whereas MTL is not available in EBMs or in any major boosted-tree package. In NAMs, the composability of neural nets makes it easy to train multiple subnets per feature. The model can learn task-specific weights over these subnets to allow sharing of subnets (shape functions) across tasks while also allowing subnets to differentiate between tasks as needed. However, it is unclear how to implement MTL in EBMs and possibly requires changes to both the backfitting procedure and the information gain rule in decision trees. Figure 8 shows a multitask NAM architecture that can jointly learn different feature representations for each task while preserving the intelligibility and modularity of NAMs. As we show, this can benefit both accuracy and interpretability. We first demonstrate multitask NAMs on a synthetic dataset before showing their utility on a multitask formulation of the COMPAS recidivism prediction dataset.

**Multitask NAM Architecture**. The multitask architecture is identical to that of single task NAMs except that each feature is associated with multiple subnets and the model jointly learns a task-specific weighted sum over their outputs that determines the shape function for each feature and task. The outputs corresponding to each task are summed and a bias is added to obtain the final prediction score. The number of subnets does not need to be the same as the number of tasks — the number of subnets can be less than, equal to, or even more than the number of tasks. Although the shape plot for each task is a linear combination of the shape plots learned by each subnet for that feature, this generates a single unique shape plot for each task and there is no need to examine what has been learned by the individual subnets for interpreting multitask NAMs.

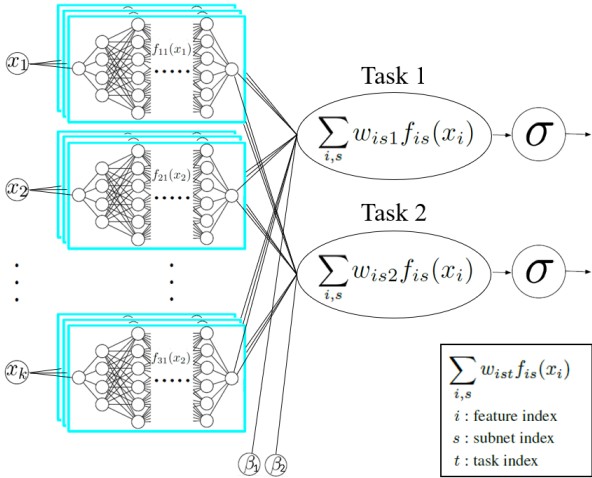

Figure 8: **Multitask NAM architecture** for binary classification. Multiple subnets are trained on each input feature and weighted sums are learned over the subnets.

### 4.2.1 Experiments on Synthetic Multitask Data

Multitask models often show improvement over single task learning when tasks are similar to each other and training data is limited. We construct a synthetic dataset that showcases the benefit of multitask learning in NAMs and demonstrates their ability to learn task-specific shape plots when needed. We define 6 related tasks, each a function of three variables. All 6 tasks are the same function of variables $x_0$ and $x_1$, and differ only in the function applied to $x_2$:

$$Task_0 = f(x_0) + g(x_1) + h(x_2) \qquad Task_1 = f(x_0) + g(x_1) + i(x_2)$$
$$Task_2 = f(x_0) + g(x_1) - h(x_2) \qquad Task_3 = f(x_0) + g(x_1) - i(x_2)$$
$$Task_4 = f(x_0) + g(x_1) + (h(x_2) + i(x_2)) \qquad Task_5 = f(x_0) + g(x_1) - (h(x_2) + i(x_2))$$

Functions $f(x_0)$, $g(x_1)$, $h(x_2)$ and $i(x_2)$ are as follows:

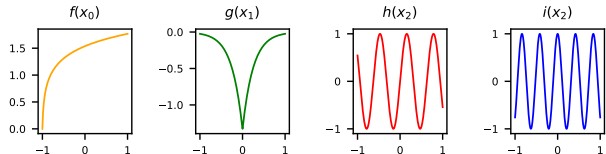

A NAM with two subnets per feature can model every function of $x_2$ by learning two subnets, one for $h(x_2)$ and one for $i(x_2)$ and assigning appropriate weights to the output of each. Because we would not know this in advance with real data, we use 6 subnets so that each of the 6 tasks (outputs) could, if needed, learn independent shape functions. We train models on 2,500 training examples, evaluate

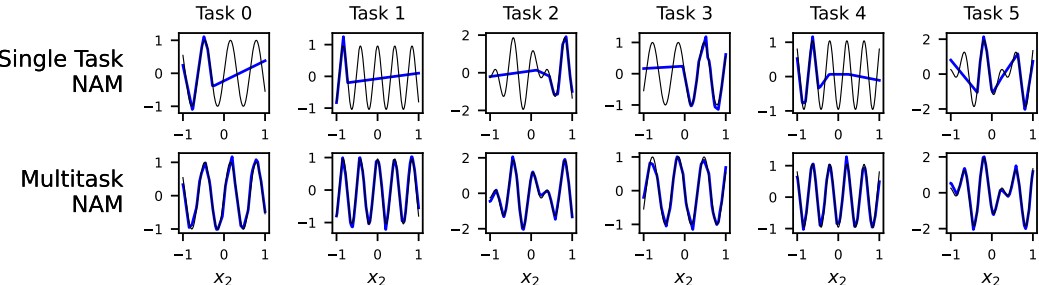

Figure 9: **Single and Multitask NAM shape plots** for $x_2$ from a typical (median) run of each task. The learned shape function is blue; the generator function is black. See A.8.2 for details of the generator functions.

Table 2: MSE for STL and MTL NAMs on synthetic data. Average of 20 runs. Lower MSEs are better.

| Model | Task 0 | Task 1 | Task 2 | Task 3 | Task 4 | Task 5 | Mean |
|---|---|---|---|---|---|---|---|
| Single Task NAM | 0.965 | 1.116 | 1.347 | 0.944 | 1.058 | 1.066 | 1.083 |
| Multitask NAM | 0.710 | 0.715 | 0.709 | 0.711 | 0.717 | 0.709 | 0.712 |

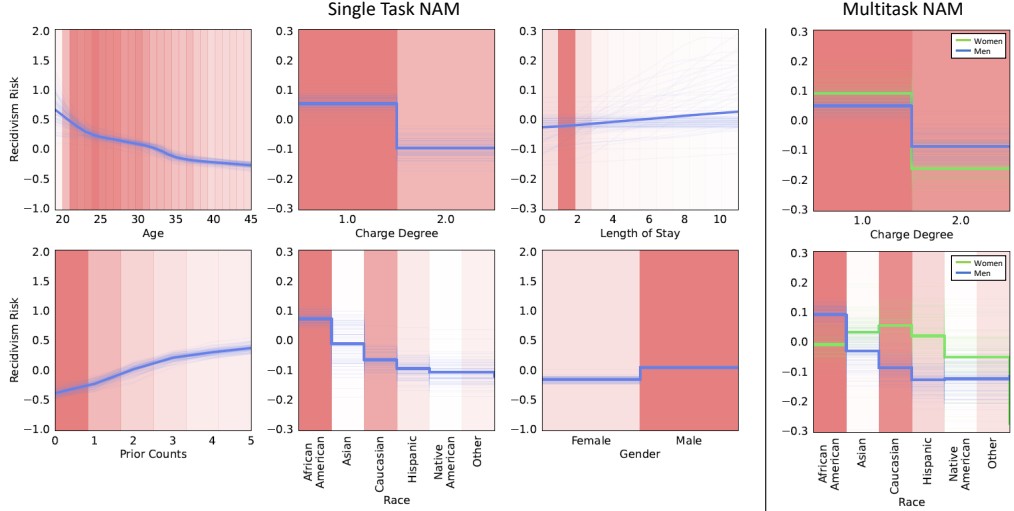

Figure 10: **Single and Multitask COMPAS Recidivism Prediction**. Plots in the left column show the shape functions for each input feature learned by an ensemble of 100 single task NAMs. Thin blue lines represent shape functions for individual members of the ensemble. Pink bars represent the normalized data density for each feature. Plots in the right column show the Race and Charge degree shape plots for an ensemble of 100 multitask NAMS, with the Women task shown in green, and the Men task in blue.

them on a test set of 10,000 examples, and average the results over 20 trials. Also, we ensured that each subnet has enough parameters to easily learn the necessary feature shape plots. So MTL is not doing better than STL because STL has inadequate capacity and MTL has more capacity.

Table 2 shows that on average across all tasks, multitask NAMs achieve mean squared error 34% lower than single task NAMs, and at least 25% lower on each individual task. In all 120 trials of the 6 tasks combined, MTL achieved a better score than STL on 119 of the 120 trials. Figure 9 shows the shape plots learned by median runs of STL and MTL for the functions of $x_2$ that vary among tasks. Furthermore, we illustrate that a multi-task NAM is as interpretable as a single task NAM by plotting the multi-task NAM predictions on the 3 input features for each of the tasks in Figure A.6.

### 4.2.2 Single and Multitask COMPAS Recidivism Prediction

COMPAS is a proprietary score developed to predict recidivism risk, which is used to inform bail, sentencing and parole decisions and has been the subject of scrutiny for racial bias [1, 8, 42]. In 2016, ProPublica released recidivism data [30] on defendants in Broward County, Florida.

**Single Task Recidivism Prediction:** First, we ask whether this dataset is biased using the transparency of single-task NAMs. Figure 10 shows the learned single-task NAM which is as accurate as black-box models on this dataset (see Table 1). The shape function for race indicates that the learned NAM may be racially biased: Black defendants are predicted to be higher risk for reoffending than white or Asian defendants. This suggests that the recidivism data may be racially-biased. The modularity of NAMs makes it easy to correct this bias by simply removing the contributions learned from the race attribute by zeroing out its mean-centered graph in the learned NAM. Although this would drop the AUC score as we are removing a discriminative feature, it may be a more fair model to use for making bail decisions. It is important to keep potentially offending attributes in the model during training so that the bias can be detected and then removed after training. If the offending variables are eliminated before training, it makes debiasing the model more difficult: if the offending attributes are correlated with other training attributes, the bias is likely to spread to those attributes [3]. The transparency and modularity of NAMs allows one to detect unanticipated biases in data and makes it easier to correct the bias in the learned model.

Table 3: ROC AUC for multitask and single task NAMs on COMPAS dataset, broken down by gender. Each cell contains the mean AUC ± one standard deviation obtained via 5-fold cross validation. Higher AUCs are better.

| Model | COMPAS Women | COMPAS Men | COMPAS Combined |
|---|---|---|---|
| Single Task NAM | $0.716 \pm 0.026$ | $0.735 \pm 0.009$ | $0.737 \pm 0.010$ |
| Multitask NAM | $0.723 \pm 0.019$ | $0.737 \pm 0.009$ | $0.739 \pm 0.010$ |

**Multitask Recidivism Prediction:** In some settings multitask learning can increase accuracy and intelligibility by learning task-specific shape plots that expose task-specific patterns in the data that would not be learned by single task learning. We reformulate COMPAS as a multitask problem where recidivism prediction for men and women are treated as separate tasks on a NAM with two outputs. Indeed, we find that a multitask NAM reveals different relationships between race, charge degree, and recidivism risk for men and women while achieving slightly higher overall accuracy.

The right column of Figure 10 displays a selection of shape plots learned for a multitask NAM trained on the same data as the single task NAM but with Male and Female as separate output tasks. (The remaining MTL shape plots are similar for the two genders, reinforcing that these are strongly related tasks, but we omit them for brevity.) The race shape plot in the multitask NAM shows a different pattern of racial bias for each gender. The curve for men looks similar to that of the single task NAM (which is expected because men make up 81% of the data), but the curve for women suggests that recidivism risk is lower for Black women and higher for Caucasian and Hispanic women than for their male counterparts. The multitask shape plots also reveal that charge degree is almost twice as important for women as it is for men. The straightforward extension of NAMs to the multitask setting offers a useful modelling technique not currently available with tree-based GAMs.

## 5 Related Work

Generalized Additive Neural Networks (GANNs) [29] are somewhat similar to the NAMs we propose here. Like NAMs, GANNs used a restriction in the neural net architecture to force it to learn additive functions of individual features. GANNs, however, predate deep learning and use a single hidden layer with typically only 1-5 hidden units. Furthermore, GANNs did not use backpropagation [37], required human-in-the-loop evaluation and were not successful in training accurate or scalable GAMs with neural nets. See Section A.7 for a more detailed overview of GANNs.

In contrast, NAMs in this paper benefit from the advances in deep learning. They use a large number of hidden units and multiple hidden layers per input feature subnet to allow more complex, more accurate shape functions to be learned. Furthermore, NAMs use novel ExU hidden units to allow subnets to learn the more non-linear functions often required for accurate additive models, and then form an ensemble of these nets to provide uncertainty estimates, further improve accuracy and reduce the high-variance that can result from encouraging the model to learn highly non-linear functions.

Prior to NAMs, the state-of-the-art in high-accuracy, interpretable generalized additive models [12, 14] are the GAM [23] and GA$^2$M [24] based on regularized boosted decision trees which were successfully applied to healthcare datasets [5]. We compare the accuracy of NAMs to these models in Section 3. We note that pairwise interactions, similar to GA$^2$Ms, can be easily added to NAMs – GA$^2$Ms use a heuristic to compute the importance of each pairwise interaction by fitting residual from first-order terms and select the k ($\leq 10$) most important interactions. We don't consider such interactions to keep the paper focused on additive modeling with neural nets.

## 6 Conclusion and Future Work

We present Neural Additive Models (NAMs), which combine the inherent interpretability of GAMs with the expressivity of DNNs, opening the door for other advances in interpretability in deep learning. NAMs are competitive in accuracy to GAMs and accurate alternatives to prevalent interpretable models (*e.g.,* shallow trees) while being more easily extendable than existing GAMs due to their differentiability and composability.

A promising direction for future work is improving the expressivity of NAMs by incorporating higher-order feature interactions. While such interactions may result in more expressivity, they might worsen the intelligibility of the learned NAM. Thus, finding a small number of crucial interactions seems important for more expressive yet intelligible NAMs. Another interesting avenue is developing

better activation functions or feature representations for easily expressing complex functions using NAMs. For example, fourier features [43] have been shown to be highly effective for learning high frequency functions via neural networks and might be useful for training expressive NAMs.

Extending and applying NAMs beyond tabular data to more complex tasks with high-dimensional inputs, such as computer vision and language understanding, is an exciting avenue for future work. While NAMs only use some of the expressivity of DNNs, one can imagine using NAMs in a real-world pipeline where intelligibility is required for decision making from representation [36] (*e.g.,* features learned from images, speech etc). Much of the existing interpretability work in deep learning focuses on making learned representations interpretable. Also, NAMs can be used for interpretability across multiple raw features (*e.g.,* multimodal inputs) where interpretability within a NAM network can utilize existing interpretability methods in ML – recently CNN-LSTM based extension of NAMs have already been developed for genomics [40] where the input to each NAM network was a one-hot encoded DNA sequence (passed as an image). Overall, we believe that NAMs are likely to broaden the use of inherently interpretable models in the deep learning community.

## Broader Impact

Interpretability in AI systems might be desirable or necessary for various reasons – see [44] for an overview; we discuss some of them in the context of NAMs below:

- **Safeguarding against bias**: NAMs can check whether training data is used in ways that result in bias or discriminatory outcomes and can be easily corrected for bias to yield possibly more fair models – *e.g.,* Section 4.2.2 demonstrates this utility for recidivism risk prediction.

- **Improving AI system design**: NAMs allow developers to interrogate why it behaved in a certain way (*e.g.,* tracking system malfunctions), and develop improvements – Section A.1 shows that NAMs can explain seemingly anomalous results in healthcare as well as uncover problems that might put some kinds of patients at risk and need correction before deploying the system.

- **Adhering to regulatory standards or policy requirements**: Interpretability of NAMs can be important in enforcing legal rights surrounding a system – *e.g.,* credit scores in the United States, have a well-established "right to explanation". NAMs can also enable individuals to contest model outputs, *e.g.,* challenging an unsuccessful loan application, based on the interpretations provided by NAMs for a specific decision (Figure 5).

- **Assessing risk, robustness, and vulnerability**: This can be particularly important if an AI system is deployed in a new environment, where we cannot be sure of its effectiveness – *e.g.,* NAMs for fraud detection (Section A.2.2) can be analyzed to understand the risks involved or how it might fail before deploying it to unseen customers.

- **Giving users confidence in the system**: Interpretations from NAMs might provide users confidence that it works as intended – *e.g.,* expensive house prices near metropolitan areas such as San Francisco, as predicted by NAMs (Figure 6), is expected for a trustworthy model.

- **Data-driven scientific discovery**: NAMs can be applied in natural sciences – *e.g.,* ecology [12], medicine [13], astronomy [15] *etc.* – to obtain novel scientific insights and discoveries from observational or simulated data [34, 45] while remaining scalable to the ever-increasing data.

There are also pitfalls associated with interpretability methods – NAMs are no exception. Different contexts give rise to different interpretability needs – *e.g.,* public have different expectations of systems used in healthcare *vs.* recruitment [19]. Furthermore, AI system designs often need to balance competing demands – *e.g.,* to optimize the accuracy of a system or ensure fairness (NAMs for making bail decisions with race feature "removed" may be less accurate but more fair). In many critical decision-making areas – *e.g.,* healthcare, justice, and public services – complex processes have developed over time to provide safeguards, audit functions, or other forms of accountability. NAMs may therefore be only the first step in creating trustworthy systems. Those developing NAMs must consider how their use fits in the wider socio-technical context of its deployment

## Acknowledgments

We would like to thank Kevin Swersky for reviewing an early draft of the paper. We also thank Sarah Tan for providing us with pre-processed versions of some of the datasets used in the paper. RA would also like to thank Marlos C. Machado and Marc G. Bellemare for helpful discussions.

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
