# A  Supplementary Material for Neural Additive Models

## A.1  NAMs on MIMIC-II: Mortality Prediction in ICUs

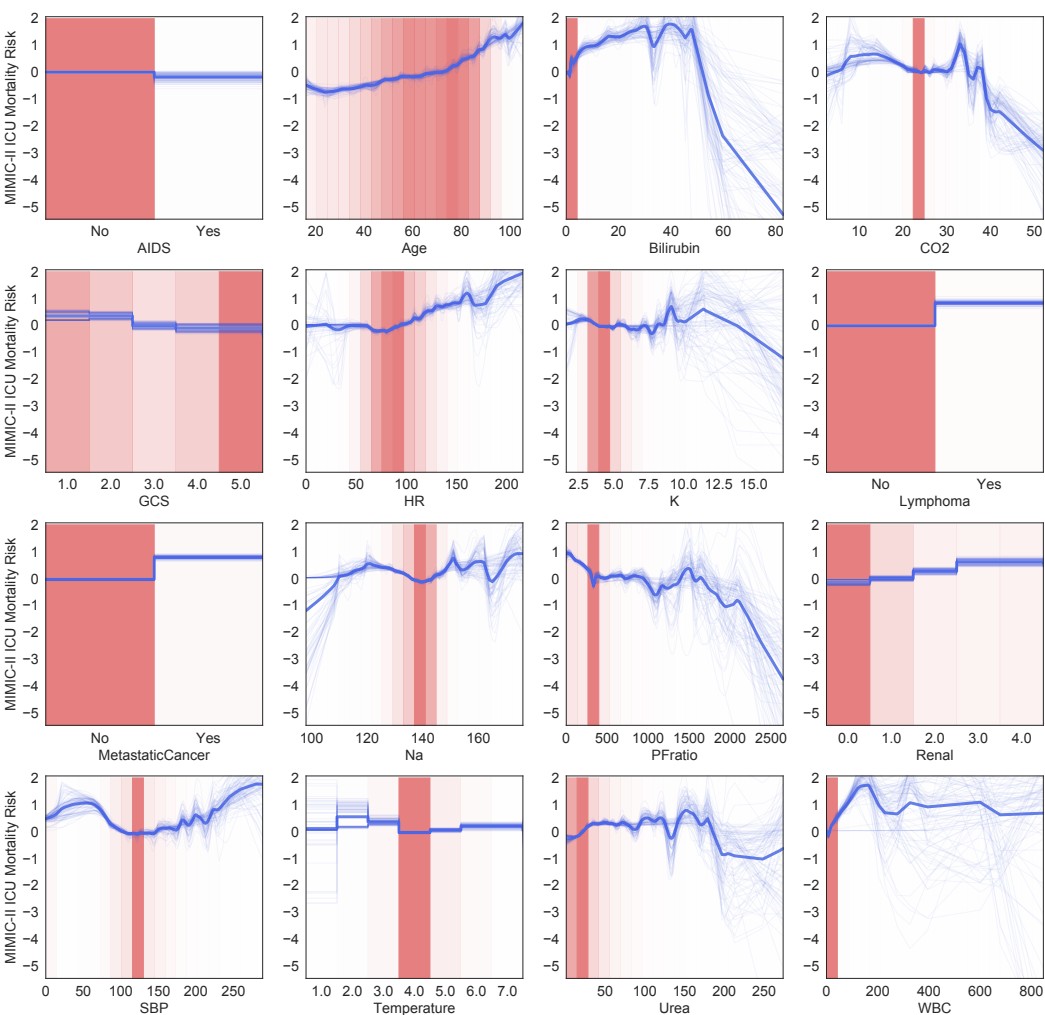

Figure A.1: **MIMIC-II ICU Mortality**. NAM shape functions learned on the MIMIC-II dataset to predict mortality risk using medical features (shown on the $x$-axis) collected during the stay in the ICU. Low values on the $y$-axis indicates a low risk of mortality.

Figure A.1 shows 16 of the shape functions learned by the NAM for the MIMIC-II dataset [38] to predict mortality in intensive care unit (ICUs). (The $17^{th}$ graph for Admission Type is flat and we omit it to save space.) The plot for HIV/AIDS shows that patients with AIDS have less risk of ICU mortality. While this might seem counter-intuitive, we confirmed with doctors that this is probably correct: among the various reasons why one might be admitted to the ICU, AIDS is a relatively treatable illness and is one of the less risky reasons for ICU admission. In other words, being admitted to the ICU for AIDS suggests that the patient was not admitted for another riskier condition, and thus the model correctly predicts that the patient is at less risk than other non-AIDS patients.

The shape plot for Age shows, as expected, that mortality risk tends to increase with age, with the most rapid rise in risk happening above age 80. There is detail in the graph that is interesting and warrants further study such as the small increase in risk at ages 18 and 19, and the bump in risk at age 90 — jumps in risk that happen at round numbers are often due to social effects.

The shape plot for Bilirubin (a by product of the breakdown of red blood cells) shows that risk is low for normal levels below 2-3, and rises significantly for levels above 15-20, until risk drops again above 50. There is also a surprising drop in risk near 35 that requires further investigation. We believe

the drop in risk above 50 is because patients above 50 begin to receive dialysis and other critical care and these treatments are very effective. The drop in risk that occurs for Urea above 175 is also likely due to dialysis.

The plot for the Glasgow Coma Index (GCS) is monotone decreasing as would be expected: higher GCS indicates less severe coma. Note that NAMs are not biased to learn monotone functions such as this and the shape of the plot is driven by the data. The NAM also learns a monotone increasing shape plot for risk as a function of renal function. This, too, is as expected: 0.0 codes for normal renal function and 4.0 indicates severe renal failure.

The NAM has learned that risk is least for normal heart rate (HR) in the range 60-80, and that risk rises as heart rate climbs above 100. Also, as expected, both Lymphoma and Metastatic Cancer increase mortality risk. The CO2 graph shows low risk for the normal range 22-24. There is an interesting drop in risk at CO2 equal to 37 (the dip between the peaks at 33 and 39) that warrants further investigation.

The shape plot for PFratio (a measure of the effectiveness of converting O2 in air to O2 in blood) shows a drop at PFratio = 332 which upon further inspection is due to missing values in PFratio being imputed with the mean: because most patients do not have their PFratio measured, the highest density of patients are actually missing their PFratio which was then imputed with the mean value of 332. One way to detect that imputed missing values are responsible for a dip (or rise) in a shape plot is when risk at the mean value of the attribute suddenly drops (or rises) to a risk level similar to what the model learns for patients who are considered normal/healthy in this dimension: the jump happens at the mean when imputation is done with the mean value, and the level jumps towards the risk of normal healthy patients because often the variable was not measured because the patients were considered normal (for this attribute), a medical assessment which is often correct.

Normal temperature is coded as 4.0 on the temperature plot, and risk rises for abnormal temperature above and below this value. It's not clear if the non-monotone risk for hypothermic patients with temperatures 1 or 2 is due to variance, an unknown problem with the data, or an unexplained but real effect in the training signals and warrants further investigation. Similarly, the shape plot for Systolic Blood Pressure (SBP) shows lowest risk for normal SBP near 120, with risk rising for abnormally elevated or depressed SBP. The jumps in risk that happen at 175, 200, and 225 are probably due to treatments that doctors start to apply at these levels: the risk rises to left of these thresholds as SBP rises to more dangerous levels but before the treatment threshold is reached, and then drops a little to the right of these thresholds when most patients above the treatment threshold are receiving a more aggressive treatment that is effective at lowering their risk.

**Discussion**. In summary, most of what the NAM has learned appears to be consistent with medical knowledge, though a few details on some of the graphs (e.g., the increase in risk for young patients, and the drop in risk for patients with Bilirubin near 35) require further investigation. NAMs are attractive models because they often are very accurate, while remaining interpretable, and if a detail in some graph is found to be incorrect, the model can be edited by re-drawing the graph. However, NAMs (like all GAMs), are not causal models. Although the shape plots can be informative, and can help uncover problems with the data that might need correction before deploying the models, the plots do not tell us *why* the model learned what it did, or what the impact of intervention (*e.g.,* actively lowering a patient's fever or blood pressure) would be. The shape plots do, however, tell us *exactly* how the model makes its predictions.

## A.2   Intelligibility of NAMs on other datasets

### A.2.1   FICO Score: Understanding Individual Predictions on Credit Scores

The FICO score is a widely used proprietary credit score to determine credit worthiness for loans in the United States. The FICO dataset [9] is comprised of real-world anonymized credit applications made by customers and their assigned FICO Score, based on their credit report information. We visualize the feature contributions of a NAM trained using the FICO dataset (see Figure A.4 in appendix) for two applicants (Table A.3) with low and high scores respectively.

Figure 5 shows that the most important features for the high scoring applicant are (1) Average Months on File and (2) Net Fraction Revolving Burden (*i.e.,* percentage of credit limit used) which take the value 235 months and 0% respectively. This makes sense, as generally, the longer a person's credit history, the better it is for their credit score. Although there is a strong inverse correlation between

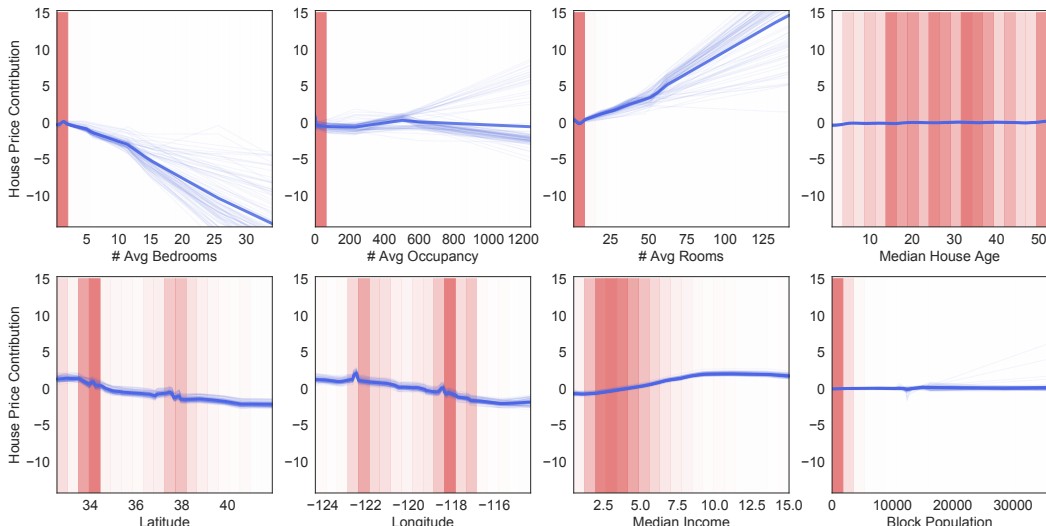

Figure A.2: **California Housing**. Graphs learned by NAMs trained to predict house prices (regression) on the California Housing dataset. These plots show the individual shape functions learned by an ensemble of hundred NAMs for each input feature as well as the data density. The thin blue lines represents different shape functions from the ensemble to show the agreement of the members of the ensemble. The pink bars represent the normalized data density for each feature. The darker the bar the more data there is with that value.

Net Fraction Revolving Burden and the score, it is positively correlated for small values ($< 10$). This means that making use of some credit increases your credit score, but using too much of it is bad. We are confident in this interpretation because most of the data density is in small values, and each NAM in the ensemble displays a similar shape function (Figure A.4). For the low scoring applicant, the main factors are (1) Total Number of Trades[4] and (2) Net Fraction Installment Burden (Installment balance divided by original loan amount) which take the values 57 and 68% respectively. This applicant used their credit quite frequently and has a large burden, thus resulting in a low score.

### A.2.2   Credit Fraud: Financial Fraud Detection [Classification]

This is a large dataset [7] containing 284,807 transactions made by European credit cardholders where the task is to predict whether a given transaction is fraudulent or not. It is highly unbalanced and contains only 492 frauds (0.172% of the entire dataset) of all transactions. Table 1 shows that on this dataset, NAMs outperform EBMs and perform comparably to the XGBoost baseline. This shows the benefit of using NAMs instead of tree-based GAMs and suggests that NAMs can provide highly accurate and intelligible models on large datasets. NAMs using ExU units perform much better compared to NAMs with standard DNNs (AUC $\approx 0.974$).

### A.2.3   California Housing: Predicting Housing Prices [Regression]

California Housing dataset [27] is a canonical machine learning dataset where the task is to predict the median price of houses (in million dollars) in each California district. The learned NAM considers the median income as well as the house location (latitude, longitude) as the most important features (we omit the other six graphs to save space, see Figure A.2). As shown by Figure 6, the house prices increase linearly with median income in high data density regions. Furthermore, the graph for longitude shows sharp jumps in price prediction around 122.5°W and 118.5°W which roughly correspond to San Francisco and Los Angeles respectively.

### A.3   Regularization and Training

ExU units encourage learning highly jagged curves, however, most realistic shape functions tend to be smooth with large jumps at only a few points. To avoid overfitting, we use the following regularization techniques:

- **Dropout** [41]: It regularizes ExUs in each feature net, allowing them to learn smooth functions while being able to represent jumps (Figure 3).

---

[4]Credit trades refer to any agreement between a lending agency and consumers.

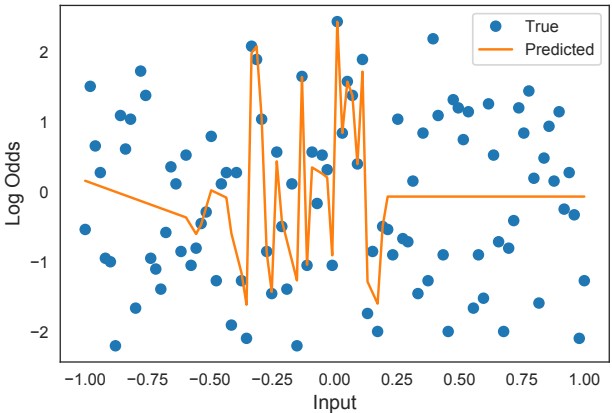

Figure A.3: **Toy classification**: Deep neural network with 3 hidden layers of size 64, 64 and 32 respectively with ReLU activation and Xavier initialization trained for 10,000 epochs on toy classification dataset described in Section 2. We use a batch size of 1024 with the Adam optimizer and a learning rate decay of 0.995 every epoch. The learning rate was tuned in [1e-3, 1e-1] and we show the results with the best learning rate.

- **Weight decay**: This is done by penalizing the L2 norm of weights in each feature net.
- **Output Penalty**: We penalize the L2 norm of the prediction of each feature net, so that its contribution stays close to zero unless evident otherwise from the data.
- **Feature Dropout**: We also drop out individual feature networks during training. When there are correlated input features, an additive model can possibly learn multiple explanations by shifting contributions across these features. This term encourages NAMs to spread out those contributions.

**Training**. Let $\mathcal{D} = \{(\mathbf{x}^{(i)}, y^{(i)})\}_{i=1}^{N}$ be a training dataset of size $N$, where each input $\mathbf{x} = (x_1, x_2, \ldots, x_K)$ contains $K$ features and $y$ is the target variable. In this work, we train NAMs using the loss $\mathcal{L}(\theta)$ given by

$$\mathcal{L}(\theta) = \mathbb{E}_{x,y\sim\mathcal{D}}\big[l(x,y;\theta) + \lambda_1 \eta(x;\theta)\big] + \lambda_2 \gamma(\theta) \tag{3}$$

where $\eta(x;\theta) = \frac{1}{K}\sum_x \sum_k (f_k^\theta(x_k))^2$ is the output penalty, $\gamma(\theta)$ is the weight decay and $f_k^\theta$ is the feature network for the $k^{\text{th}}$ feature. Each individual network is also regularized using feature dropout and dropout with coefficients $\lambda_3$ and $\lambda_4$ respectively. $l(x,y;\theta)$ is the task dependent loss function. We use the cross-entropy loss for binary classification:

$$l(x,y;\theta) = -y\log(p_\theta(x)) - (1-y)\log(1 - p_\theta(x)),$$

where $p_\theta(x) = \sigma\big(\beta^\theta + \sum_{k=1}^{K} f_k^\theta(x_k)\big)$ and mean squared error (MSE) for regression:

$$l(x,y;\theta) = \big(\beta^\theta + \sum_{k=1}^{K} f_k^\theta(x_k) - y\big)^2$$

### A.4 Some practical considerations when using NAMs

How NAMs performs when the underlying features are additive (*i.e.,* no non-linearities)? We empirically observed that the NAM MLPs do end up approximately recovering the linear functions. That said, the inductive bias of NAMs is toward learning non-linear functions and they might be more expensive than linear models – once a user sees that a NAM learns a linear function for a specific feature, they can try substituting that feature network with a simpler one (or non-linear one) to see if that improves generalization.

Were there instabilities when using ExUs? Surprisingly, we did not observe any instability in training dynamics (across the 4 datasets and synthetic example) and we speculate this is because any small change in weights can lead to significantly peaky function which results in huge loss on training points. Also, we used the Adam optimizer, which adapts the norm of the gradient and prevents them from exploding. Various regularization approaches including weight-regularization and dropout further stabilize the dynamics.

Should we use ExUs vs ReLUs? While a general guidance might be tricky, we hope that ExUs might help certain users to benefit more from the ability of fitting smooth functions but with large jumps at a few point. Devising better activation functions for NAMs is an open research problem.

## A.5 Experimental Details

**Training Details**. The NAM feature networks ($f_k^\theta$) are trained jointly using the Adam optimizer [20] with a batch size of 1024 for a maximum of 1000 epochs with early stopping using the validation dataset. The learning rate is annealed by a factor of 0.995 every training epoch. For all the tasks, we tune the learning rate, output penalty coefficient ($\lambda_1$), weight decay coefficient ($\lambda_2$), dropout rate ($\lambda_3$) and feature dropout rate ($\lambda_4$). For computational efficiency, we tune these hyperparameters using Bayesian optimization [11, 39] based on cross-validation performance with a single train-validation split for each fold. We used TESLA P100 GPUs for all experiments involving neural networks while CPU machines for *sklearn* or XGBoost baselines.

**Evaluation**. We perform 5-fold cross validation to evaluate the accuracy of the learned models. To measure performance, we use area under the precision-recall curve (AUC) for binary classification (as the datasets are unbalanced) and root mean-squared error (RMSE) for regression. For NAMs and DNNs, one of the 5 folds (20% data) is used as a held-out test set while the remaining 4 folds are used for training (70% data) and validation (10% data). The training and validation splits are randomly subsampled from the 4 folds and this process is repeated 20 times. For each run, the validation set is used for early stopping. For each fold, we ensemble the NAMs and DNNs trained on the 20 and EBMs on 100 train-validation splits respectively to make the prediction on the held-out test set.

## A.6 Hyperparameters

We use a batch size of 1024 with the Adam optimizer and a learning rate decay of 0.995 every epoch in our experiments for NAMs and DNNs.

**Linear Models/ Decision Trees**. We use the *sklearn* implementation [28], and tune the hyperparameters with grid search.

**EBMs**. We use the open-source implementation [26] with the parameters specified by prior work [5] for a fair comparison.

**NAMs**. We tune the dropout coefficient ($\lambda_3$) in the discrete set {0, 0.05, 0.1, 0.2, 0.3, 0.4, 0.5, 0.6, 0.7, 0.8, 0.9}, weight decay coefficient ($\lambda_2$) in the continuous interval [0.000001, 0.0001), learning rate in the interval [0.001, 0.1), feature dropout coefficient ($\lambda_4$) in the discrete set {0, 0.05, 0.1, 0.2} and output penalty coefficient ($\lambda_1$) in the interval [0.001, 0.1). Note that the weight decay is implemented as the average weight decay over the individual feature networks in NAMs. Refer to Table A.2 and Table A.1 for hyperparameters found on regression and classification datasets.

**DNNs**. We train DNNs with 10 hidden layers containing 100 units each with ReLU activation using the Adam optimizer. This architecture choice ensures that this network had the capacity to achieve perfect training accuracy on datasets used in our experiments. We use weight decay and dropout to prevent overfitting and tune hyperparameters using a similar protocol as NAMs. We tune the dropout coefficient ($\lambda_3$) in {0, 0.05, 0.1, 0.2, 0.3, 0.4, 0.5}, weight decay coefficient ($\lambda_2$) in the continuous interval [0.0000001, 0.1) and learning rate in the interval [0.001, 0.1).

Table A.1: Optimal hyperparameters found for NAMs on regression datasets. "Hidden units" shows the number of hidden layers as well as the number of neurons used in each layer for each feature network.

| Hyperparameter | FICO | Housing |
|---|---|---|
| Learning Rate | 0.0161 | 0.00674 |
| Output Penalty ($\lambda_1$) | 0.0205 | 0.001 |
| Weight Decay ($\lambda_2$) | 1.07 x $10^{-5}$ | $10^{-6}$ |
| Dropout | 0.0 | 0.0 |
| Feature Dropout | 0.0 | 0.0 |
| Num units | 64, 64, 32 | 64, 64, 32 |
| Activation | ReLU | ReLU |
| Hidden unit | Standard | Standard |

Table A.2: Optimal hyperparameters found for NAMs on classification datasets. "Hidden units" shows the number of hidden layers as well as the number of units used in each hidden layer for each feature network.

| Hyperparameter | COMPAS | MIMIC-II | Credit Fraud |
|---|---|---|---|
| Learning Rate | 0.02082 | 0.005 | 0.0157 |
| Output Penalty | 0.2078 | 0.3 | 0.0 |
| Weight Decay | 0.0 | $9.6 \times 10^{-5}$ | $4.95 \times 10^{-6}$ |
| Dropout | 0.1 | 0.2 | 0.8 |
| Feature Dropout | 0.05 | 0.0 | 0.0 |
| Num units | 64, 64, 32 | 1024 | 1024 |
| Activation | ReLU | ReLU-1 | ReLU-1 |
| Hidden unit | Standard | ExU | ExU |

## A.7 GANNs

GANNs begin with subnets containing a single hidden unit for each input feature, and use a human-in-the-loop process to add (or subtract) hidden units to the architecture based on human evaluation of plotted partial residuals. This means that the training procedure cannot be automated. In practice, the laborious manual effort required to evaluate all of the partial residual plots to decide what to do next, and then retrain the model after adding or subtracting hidden units from the architecture meant that GANN nets remained very small and simple — typically only one hidden unit per feature.

Table A.3: Feature attributes for the two individuals shown in Figure 5

| Feature | High Score Applicant | Low Score applicant |
|---|---|---|
| Months Since Oldest Trade Open | 417.0 | 174.0 |
| Months Since Most Recent Trade | 25.0 | 1.0 |
| Average Months in File | 235.0 | 66.0 |
| # Satisfactory Trades | 9.0 | 44.0 |
| # Trades 60+ Ever | 0.0 | 11.0 |
| # Trades 90+ Ever | 0.0 | 8.0 |
| % Trades Never Delinquent | 100.0 | 70.0 |
| Months Since Most Recent Delinquency | 0.0 | 3.0 |
| Max Delq/Public Records Last Year | 6.0 | 0.0 |
| Max Delinquency Ever | 6.0 | 0.0 |
| # Total Trades | 9.0 | 57.0 |
| # Trades Open in Last 12 Months | 0.0 | 5.0 |
| % Installment Trades | 22.0 | 66.0 |
| Months Since Most Recent Inquiry excluding 7 days | 0.0 | 0.0 |
| # Inquiries in Last 6 Months | 1.0 | 7.0 |
| # Inquiries in Last 6 Months excluding 7 days | 0.0 | 6.0 |
| Net Fraction Revolving Burden | 0.0 | 23.0 |
| Net Fraction Installment Burden | 0.0 | 68.0 |
| # Revolving Trades with Balance | 1.0 | 2.0 |
| Number Installment Trades with Balance | 0.0 | 5.0 |
| # Bank/Natl Trades with high utilization ratio | 0.0 | 0.0 |
| % Trades with Balance | 40.0 | 64.0 |
| Delinquent | 0.0 | 1.0 |
| Inquiry | 1.0 | 1.0 |

Table A.4: **FICO Score**. Meaning of the different attributes of the feature "Max Delq/Public Records Last Year".

| Value | Meaning |
|-------|---------|
| 0 | Derogatory comment |
| 1 | 120+ days delinquent |
| 2 | 90 days delinquent |
| 3 | 60 days delinquent |
| 4 | 30 days delinquent |
| 5,6 | Unknown delinquent |
| 7 | Current and never delinquent |
| 8,9 | All other |

## A.8 Multitask Learning

### A.8.1 Synthetic Data Generation

We used the following generator functions to produce our synthetic dataset:

$$
\begin{aligned}
f(x_0) &= \frac{1}{3} \log 100 x_0 + 101 \\
g(x_1) &= -\frac{4}{3} e^{-4|x_1|} \\
h(x_2) &= \sin{(10 x_2)} \\
i(x_2) &= \cos{(15 x_2)}
\end{aligned}
\tag{4}
$$

Noise sampled from $N(0, \frac{5}{6})$ was added to the target for each task. We will provide our generation code for others who are interested in using this dataset.

### A.8.2 Gains from multi-task NAMs

We also ran control experiments where we provide multiple subnets for each feature and each task for single-task learning (STL), and this does sometimes improve test accuracy marginally for STL. However, this still performed worse than multi-task NAMs as they are able to make use of samples across multiple tasks to learn a common function but the single-task NAM can't share samples and don't have access to enough data for learning individual shape functions.

### A.8.3 Shape Plots for All Synthetic Features

As shown in Figure A.6, we include here shape plots for all features in the synthetic data for both single and multitask NAMs. The MTL results represent a single model trained on all 6 tasks. In each case, it models the shape functions for every feature and the target with high accuracy. By contrast, the STL for $Task0$, struggles to fit the data for $x_2$ and achieves low accuracy on the target in this regime of noise and training set size.

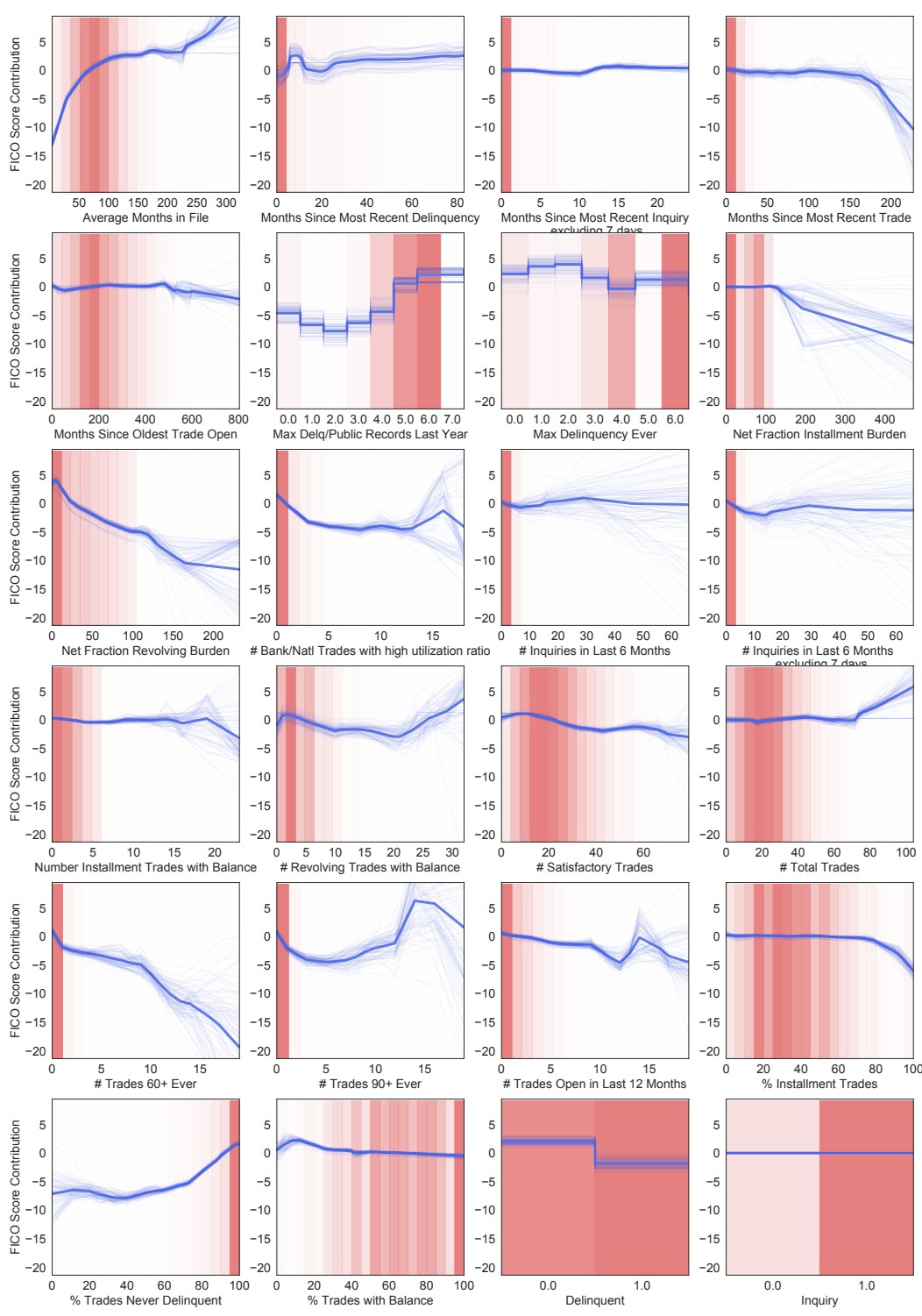

Figure A.4: **FICO Score Prediction**. Graphs learned by NAMs trained to predict FICO scores (regression) based on their credit report information. These graphs can be interpreted easily, *e.g.,* the second last graph in the bottom row shows that being delinquent on your payments decreases your credit score.

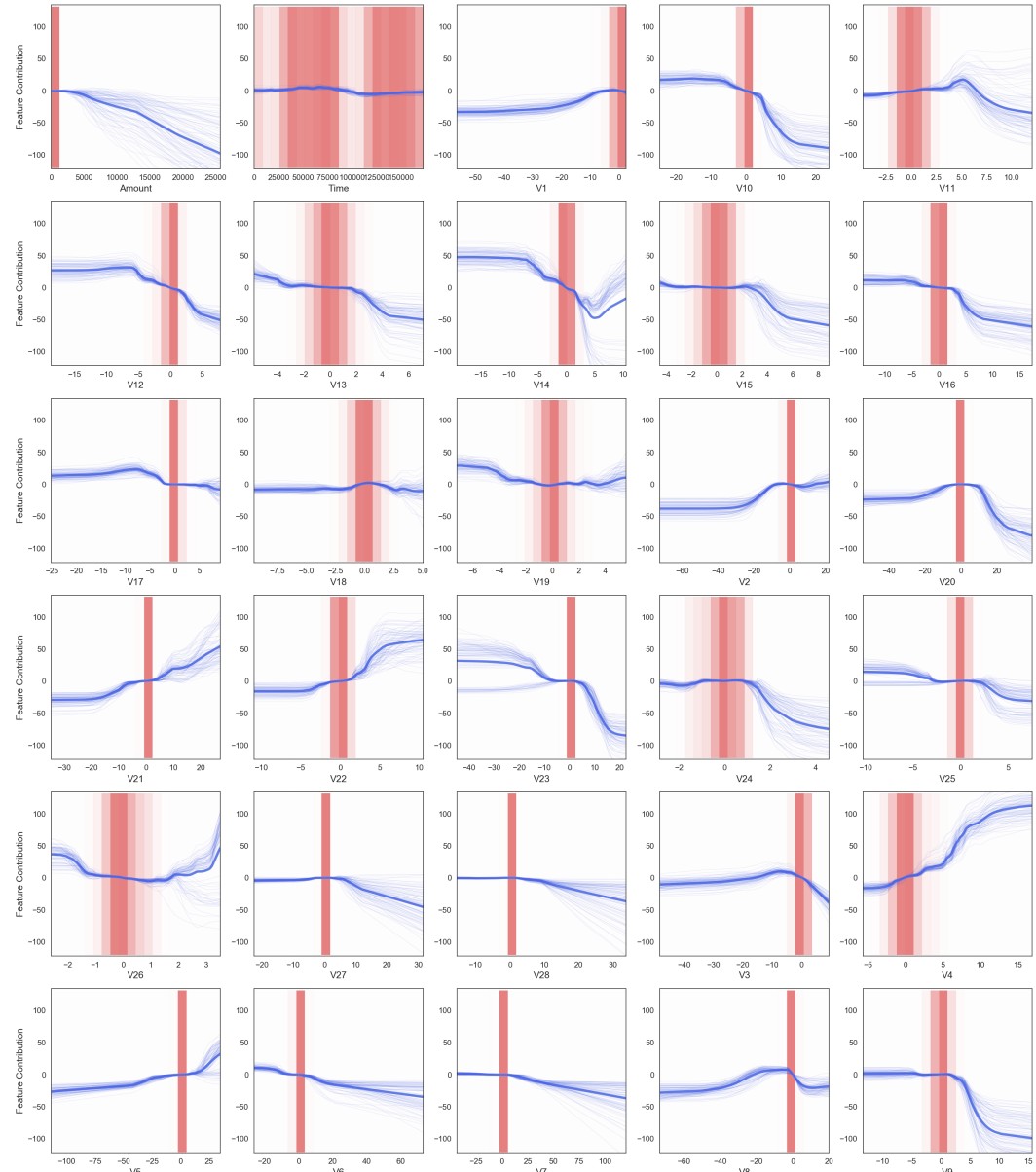

Figure A.5: **Credit Fraud Detection**: Graphs learned by NAMs with ExU units on this large classification dataset. The task is to predict credit fraud where the class variable takes value 1 in case of fraud and 0 otherwise using a large dataset of credit card transactions. The dataset only contains only numerical input variables which are the result of a PCA transformation except the features 'Time' and 'Amount'. Unfortunately, due to confidentiality issues, the original features are not provided in the dataset.

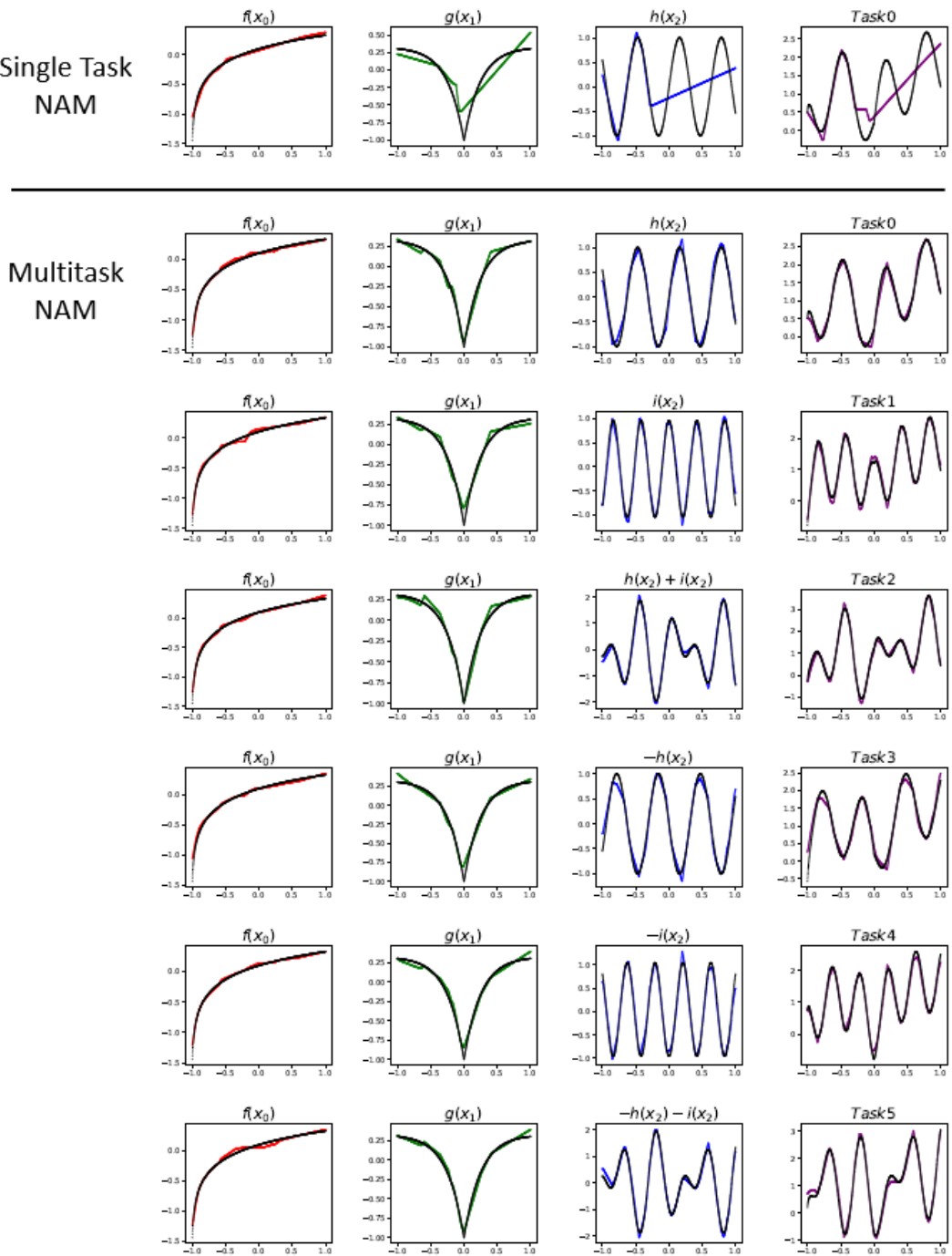

Figure A.6: **Single and Multitask NAMs trained on synthetic data**: Shape plots for all synthetic features for a typical (median) run of single and multitask NAMs. The colored lines represent learned shape functions for each feature and the black line represents the generator function.