# OpenReview forum: "Neural Additive Models: Interpretable Machine Learning with Neural Nets"
_NeurIPS.cc/2021/Conference — NeurIPS 2021 Spotlight_

### Official Review · Reviewer_qyF4 · 2021-07-13

**Rating:** 8
**Confidence:** 4

**Summary:**

This paper proposes Neural Additive Models (NAMs), a linear combination of neural networks that can be used as a highly accurate and interpretable model for tabular data. In order to account for the potentially detrimental bias of NNs to be smooth, the authors introduce an exp-centered hidden unit which enables the NN to better fit jagged shaped function, and compare this NAM to other tree-based and DNN based models. NAMs are able to achieve comparable accuracy to tree-based models with the added advantage of being easily adapted to the multitask setting, which can be helpful in increasing accuracy, discovering bias, constructing more helpful models for fields which need interpretability like healthcare.

**Limitations And Societal Impact:**

The authors addressed how this method may not be suitable for certain applications, as they may need a different type of interpretability. One other thing I would think would be an issue is the amount of resources needed/throughput, although I have not benchmarked this myself.

**Main Review:**

This paper is incredibly well written, easy to follow, and has some of the best visualizations I have seen. The motivation and core method are clear and the analysis was quite thorough and seems like it could be easily reproduced. Furthermore, the applications to things like covid-19 and COMPAS are well thought out and have impressive results. Overall a very fun read and seems particularly well suited to the NerulIPS audience.

My only concerns are relatively small:

(1) In section 2.1 the authors mention that they altered the weight initialization to better fit jagged curves; I would love to see a plot showing the difference in curves with this new initialization as well as the traditional weight initialization (similar to figure 2). I looked in A.3 but that section seems to be empty.

(2) While the plots in section 2.1 are convincing that these ExU networks are better for jagged curves, it would be nice to add them as another benchmark in section 3

Other small nits:

* Units of median income are missing in Fig 6
* line 327 I believe should be "the public has different..."
* a period needs to be added at the end of the conclusion

**Time Spent Reviewing:**

2

---

> ### Author Response · Authors · 2021-08-09
> **Author response to reviewer qyF4**
>
> We thank the reviewer for their valuable comments and address their minor concerns below.
>
> > **I would love to see a plot showing the difference in curves with this new initialization and the traditional weight initialization (similar to figure 2).**
>
> The results from our experiments on the synthetic example indicated that standard weight initialization ends up fitting smoother curves and we’ll add these comparisons in the appendix. The figure in Appendix A.3 ended up being misplaced on the next page due to formatting issues but it shows that even deeper networks with relu activations couldn’t fit the synthetic example.
>
> > **While the plots in section 2.1 are convincing that these ExU networks are better for jagged curves, it would be nice to add them as another benchmark in section 3.**
>
> This is a good suggestion – this would help show the efficacy of ExUs.

---

### Official Review · Reviewer_Jr31 · 2021-07-16

**Rating:** 7
**Confidence:** 3

**Summary:**

In this paper, the authors introduce a new generalized additive model, called neural additive models, which use neural networks to learn a non-linear transformation of each input feature, independently. Along the way, they introduce a new activation function, called ExU, which enables a highly sensitive weights that can fit very large, seemingly noisy, changes of a given function. They demonstrate that NAM performs as well as other GAMs, such as EBMs, as well as black box models, such as DNNs and XGBoost. The major  benefit of NAM is that, similar to EBMs, they are intrinsically interpretable simply by visualizing the learned non-linear function for each input feature.  They compare the performance of these methods across several tasks. The paper is written very clearly with easy to follow examples.

**Limitations And Societal Impact:**

Yes.

**Main Review:**

Overall, I found the proposed NAM very interesting and a worthwhile contribution to the toolbox of GAMs. The new activation, i.e. ExU, is also quite intriguing. In this regards, I think that the paper has legs to stand on. The work is a simple extension, using neural networks to learn a non-linear transformation of each input feature independently. It is thus learning a suitable non-linearity for each input feature such that its additive contribution is predictive of the task at hand. Instead of using a preset non-linear function, NAMs use an MLP to learn one. It is very clear how this approach is different from previous contributions. A major limitation to this study was that there was no clear example that highlights the benefits of NAM over previous methods, either in terms of performance or interpretability. Below, I highlight my main concerns:

- The argument that a ExU can learn jagged shape functions, whereas the same network with ReLU activations learns something smoother is clearly demonstrated for a given network. However, a deeper network with ReLU activations should be able to fit the noisier function, right? So, the statement is really more about the ability of ExU to express noisier functions using a smaller number of parameters compared to ReLU activations.

- Many of the synthetic examples are a bit contrived. For instance, the example of learning a jagged shape function; It's actually unclear when learning this kind of jagged shape function would be required in practice. On the flip side, how well does ExU do when the underlying function is smooth? Does it over-exaggerate more easily due to its highly sensitive weights? Would a relu or smooth activation perform better in such a circumstance? If there is a big difference between these scenarios, perhaps some guidance would be beneficial.

- Building on this question, how does NAM do when the underlying features are supposed to be additive (i.e. no nonlinearities). In this case, NAMs are overkill, but do their MLPs learn learn a linear function or do they learn complex non-linear functions? If that's the case, how "interpretable" is it? I supoose it is still interpretable in a strict sense of the model, but it is very different from being reliable. A comment on this could help an end user think more deeply about these considerations.

- I do appreciate the attempt at comparing different methods across different datasets, but I think the examples don't really highlight NAMs ability to excel at tasks that previous methods struggle. The gains for each model is quite marginal. One question could be -- relative to previous methods, are NAMs robust to a wide setting of hyperparameters? Are there any benefits (beyond interpretability) with training time, hyperparameter search, inference time, scalability? If so, this should be highlighted as the performance alone is right in the pack (and EBMs are also interpretable).

- Since interpretability means so many things these days, I think there should be a clear message what is meant by interpretability here. A comparison of global function vs local function via LIME was made, but it may be nice to contrast other definitions of interpretability, such as post hoc explanations with attribution methods.

- In regards to interpretability, it is a true benefit that NAMs can just plot the learned function and one can get a sense for variations of the function. It's nice that variability is low where data is concentrated and then the uncertainty grows in data deprived regions of function space. it would also be nice to see  interpretations of other models for comparison.

- How do EMB interpretations compare? Do you draw the same conclusions? This may be beyond the scope of the paper but for black box models that were in the comparison, perhaps attribution methods, like Shap, could be sufficient for seeing whether you can draw similar conclusions (or point out weaknesses in their explanations).

- For multi-task learning, it's unclear whether connecting each MLP across tasks leads to lower interpretability  as there are now a linear combination of different non-linear functions for each feature -- not just one. The examples show the prediction of each task, but that doesn't necessarily highlight interpretability. The improved performance may be benefiting from the increased expressivity of the model and not necessarily from the fact that it is in a multi-task learning paradigm. A nice control experiment would be to keep multiple MLP modules for each feature (as if multi-tasking) but train on a single task. I suspect the model would now be able to capture the complex shapes that it was unable to with the original single-task NAM. If this experiment works, I think it would be beneficial to compare the interpretability of the quasi-multi-task vs single-task NAM.

**Time Spent Reviewing:**

5

---

> ### Author Response · Authors · 2021-08-09
> **Author response to reviewer Jr31**
>
> We thank the reviewer for their detailed comments. We address their main concerns below.
>
> > **A major limitation to this study was that there was no clear example that highlights the benefits of NAM over previous methods, either in terms of performance or interpretability. Are there any benefits (beyond interpretability) ?**
>
> We believe that the main advantage of NAMs over existing GAMs is their flexibility which allows for novel avenues of using inherently interpretable and differentiable models to settings which are highly problematic for EBMs. For example, the differentiability of NAMs allows them to train more complex interpretable models for COVID-19 (Section 4.1), which is impossible with tree-based GAMs. Similarly, the community has developed extensions of NAMs to tackle problems such as survival analysis [1], where it is not clear how tree-based GAMs could be modified.
>
> Inference time is another advantage of NAMs – NAMs can use GPUs/TPUs for training and inference while this is not possible with current state-of-the-art EBMs. Furthermore, NAMs can make use of advances in deep learning hardware. Also, EBMs require a highly parallelized implementation of boosted decision trees which is not easier to extend compared to NAMs.
>
> > **The argument that an ExU can learn jagged shape functions, whereas the same network with ReLU activations learns something smoother is clearly demonstrated for a given network. However, a deeper network with ReLU activations should be able to fit the noisier function, right? So, the statement is really more about the ability of ExU to express noisier functions using a smaller number of parameters compared to ReLU activations.**
>
> Actually, we couldn’t fit the synthetic example with a deeper neural network with even 3 hidden layers (Figure A.3). Since implicit regularization of standard neural nets trained with mini-batch gradient descent biases them towards fitting smoother functions, it is unclear if highly noisy functions can be easily fit with deeper neural networks with current techniques.
>
> > **Many of the synthetic examples are a bit contrived. For instance, the example of learning a jagged shape function; It's actually unclear when learning this kind of jagged shape function would be required in practice. On the flip side, how well does ExU do when the underlying function is smooth? Does it over-exaggerate more easily due to its highly sensitive weights? Would a relu or smooth activation perform better in such a circumstance? If there is a big difference between these scenarios, perhaps some guidance would be beneficial.**
>
> Indeed, realistic shape functions typically tend to be smooth with large jumps at only a few points. To avoid overfitting with ExUs, strong regularization is crucial (see Figure 3 and Appendix A.4) which can learn such functions. With ReLUs, we can typically fit smooth functions but they might miss some of these jumps (for example, some of the jumps we observed in MIMIC-II dataset). While a general guidance might be tricky, we hope that ExUs might help certain users to benefit more from this ability.
>
> > **Building on this question, how does NAM do when the underlying features are supposed to be additive (i.e. no nonlinearities). A comment on this could help an end user think more deeply about these considerations.**
>
> This is a great point.  We empirically observed that the NAM MLPs do end up approximately recovering the linear functions. That said, the inductive bias of NAMs is toward learning non-linear functions and they might be more expensive than linear models – once a user sees that a NAM learns a linear function for a specific feature, they can try substituting that feature network with a simpler one (or non-linear one) to see if that improves generalization. We’d include a comment on this.
>
> > **Since interpretability means so many things these days, I think there should be a clear message what is meant by interpretability here. A comparison of global function vs local function via LIME was made, but it may be ice to contrast other definitions of interpretability, such as post hoc explanations with attribution methods.**
>
> We argue that inherently interpretable glass-box models like NAMs are superior to black-box models of similar accuracy because global explanations generated for black-box models are approximate and thus not always faithful to how the black-box models computes predictions and do not provide enough detail to fully understand the model’s behavior. That said, checking whether the interpretations make sense really depends on the end user – for example,  we validated the interpretations from NAMs on the MIMIC-dataset (appendix A.1) with a doctor.
>
> > **For multi-task learning, it's unclear whether connecting each MLP across tasks leads to lower interpretability  as there are now a linear combination of different non-linear functions for each feature -- not just one. The examples show the prediction of each task, but that doesn't necessarily highlight interpretability.**
>
> Although the shape plot for each task is a linear combination of the shape plots learned by each subnet for that feature, this generates a single unique shape plot for each task and there is no need to examine what has been learned by the individual subnets.  You can think of what is learned by the subnets as a set of basis functions that are trained in parallel and which when combined with linear weighting form the shape function for each task.
>
> Figure A.6 in the appendix shows the interpretability of these multi-task models where we show the prediction on the 3 input features for each of the tasks. These plots how to interpret these predictions made by the multi-task NAMs on each of the individual tasks based on the input features and is as interpretable as a single task NAM.
>
> > **The improved performance may be benefiting from the increased expressivity of the model and not necessarily from the fact that it is in a multi-task learning paradigm. A nice control experiment would be to keep multiple MLP modules for each feature (as if multi-tasking) but train on a single task. I suspect the model would now be able to capture the complex shapes that it was unable to with the original single-task NAM. If this experiment works, I think it would be beneficial to compare the interpretability of the quasi-multi-task vs single-task NAM.**
>
> Actually, we have done the control experiments where we provide multiple subnets for each feature and each task for single-task learning (STL), and this does sometimes improve test accuracy marginally for STL. However, this still performed worse than multi-task NAMs as they are able to make use of samples across multiple tasks to learn a common function but the single-task NAM can’t share samples and don’t have access to enough data for learning individual shape functions. We omitted those results in the paper to keep the story simple, but if the reviewer wants we could add those results to the camera ready copy either in the main body of the paper or in the appendix.
>
> Also, we were careful to make sure that each subnet has enough parameters to easily learn the necessary feature shape plots. So MTL is not doing better than STL because STL has inadequate capacity and MTL has more capacity.
>
> ------------------------------------------------------------------------------------------------------------------------------------------------------------------------------------
> *We hope that our response addresses most of the concerns of the reviewer. We'd be happy to engage in further discussions.*

---

### Official Review · Reviewer_yq49 · 2021-07-17

**Rating:** 5
**Confidence:** 4

**Summary:**

This paper propose Neural Additive Model that provides more interpretability with DNNs. It learns a linear combination of neural networks that each attend to a single input feature. Through experiments, NAM is shown to be more accurate than other intelligible models such as logistic regression and explainable boosting machine (EBM), while also provides explanability with respect to the input features. A multitask learning architecture along with the single task architecture is proposed.

**Limitations And Societal Impact:**

The limitation of the proposed NAM is that it is a simple linear combination of multiple neural nets which each only attends to one feature. While it provides the intelligibility to the model when input feature is simple, its generalizability and expressiveness to more complex domains is lacking.

**Main Review:**

The proposed NAM is a DNN version of the generalized additive models (GAM). The proposed NAM allows one to exam the eact description of how NAM compute a prediction since one can sweep the input feature and see the prediction curve. The individual shape function provides the good intelligibility for the input feature. The proposed method contains certain amount of novelty in the NAM and the way to interprete the model, including the multi-task NAM architecture. However, I think the novelty of the proposed NAM is not that significant. Several experiments have been conducted to demonstrate the better accuracy compared with linear regression and the intelligible EBM. The paper also shows several examples on how to interpret the model and explain how the input features affect the performance. The paper is well-written and easy to understand.

While it is good to allow each network attend to only one feature, the limitation of the proposed NAM is also evident. The major advantage of deep learning models lie in its representation learning, which learns the non-linear combination and transformation of interleaving features, not individual features. The relationship and interaction of different features empowers the development in many areas where deep learning applies such as computer vision, natural language processing, recommendation, etc. I don't see how the proposed NAM can be more generalized to other areas, where the input feature is of high-dimension and requires more complex understanding.

Other concerns:
- The proposed exp-centered hidden units (ExU) uses the exponential function for the weight. Will this introduce instability in training? It roughly only allows the weight to be in a much smaller range, and could cause the gradient to go to nan easily.

**Time Spent Reviewing:**

2

---

> ### Author Response · Authors · 2021-08-09
> **Author response to reviewer yq49**
>
> We thank the reviewer for their feedback and respond to their concerns below.
>
> > **.. I think the novelty of the proposed NAM is not that significant.**
>
> While NAMs are simple (which is a virtue itself), their flexibility allows for novel avenues of using inherently interpretable and differentiable models in settings which are more problematic for EBMs (tree-based GAMs) or other types of GAMs. For example, the differentiability of NAMs allows them to train more complex interpretable models for COVID-19 (Section 4.1) -- this would be much more difficult with tree-based GAMs.
>
> Similarly, the community has developed extensions of neural nets to tackle problems such as survival analysis [1] that could easily be applied to NAMs, but which would be more difficult to apply to existing GAMs.  Unlike tree-based GAMs, NAMs allow the flexibility and modularity of neural nets to be applied to GAMs.
>
>  > **The relationship and interaction of different features empowers the development in many areas where deep learning..  I don't see how the proposed NAM can be more generalized to other areas, where the input feature is of high-dimension and requires more complex understanding.**
>
> While we acknowledge that NAMs only use some of the expressivity of DNNs, one can imagine using NAMs  in a real-world pipeline where intelligibility is required for decision making from representation features (learned from images, speech etc) (See [2] for details).  Much of the existing interpretability work in deep learning focuses on making learned representations interpretable.
>
> Also, NAMs can be used for interpretability across multiple raw features (multimodal inputs for example) where interpretability within a NAM network can make use of use black-box interpretability methods in ML – recently CNN-LSTM based extensions of NAMs have been applied to  genomics [3]  where the input to each NAM network was a one-hot encoded DNA sequence (passed as an image).
>
> > **The proposed exp-centered hidden units (ExU) .. will this introduce instability in training? It roughly only allows the weight to be in a much smaller range, and could cause the gradient to go to nan easily.**
>
> Surprisingly, we didn’t observe any instability in training dynamics (across the 4 datasets and synthetic example) and we speculate this is because any small change in weights can lead to significantly peaky function which results in huge loss on training points. Also, we used the Adam optimizer, which adapts the norm of the gradient and prevents them from exploding. The regularizations we use including weight-regularization and dropout further stabilize the dynamics (but weren’t necessary for stable training).  We’d add more details about it in the revision.
>
>
> [1] Utkin, L. V., Satyukov, E. D., & Konstantinov, A. V. (2021). SurvNAM: The machine learning survival model explanation. arXiv preprint arXiv:2104.08903.
>
> [2] Rudin, C., Chen, C., Chen, Z., Huang, H., Semenova, L., & Zhong, C. (2021). Interpretable machine learning: Fundamental principles and 10 grand challenges. arXiv preprint arXiv:2103.11251.
>
> [3] Srivastava, D., Aydin, B., Mazzoni, E. O., & Mahony, S. (2021). An interpretable bimodal neural network characterizes the sequence and preexisting chromatin predictors of induced transcription factor binding. Genome biology, 22(1), 1-25.
>
> ------------------------------------------------------------------------------------------------------------------------------------------------------------------------------------
> *We would appreciate it if the reviewer can confirm that their concerns had been addressed and, if so, reconsider their assessment. We’d be happy to engage in further discussions.*

---

### Official Review · Reviewer_oFHP · 2021-07-20

**Rating:** 7
**Confidence:** 2

**Summary:**

The authors propose the neural additive models which have an additive structure and which are more explainable than general neural networks. They apply this model for various tasks and applications and show their interpretability through visualizations. Its a good contribution and a reasonably well-written paper.

**Ethical Concerns:**

I have found no ethical concerns in this paper.

**Limitations And Societal Impact:**

This paper addresses a number of applications like healthcare and recidivism prediction which have a high societal impact. For example, they try to determine the features which contribute to mortality during ICU stays (using the MIMIC dataset) and the features that contribute to bias in the COMPAS dataset. Though the authors do some heuristic analysis, these results should be ideally formally validated. The authors should also address some of the limitations of their current work and potential future avenues of study.

**Main Review:**

The authors propose neural additive models, a linear combination of multiple small networks, which are trained jointly and can learn complex functions such as jagged shape functions. The claim is that because the models are modular, the predictions can be propagated back to the individual network functions and easily visualized. They show the explainability of the networks in applications like fraud detection and recidivism risk prediction.

The authors have explained the proposed structure reasonably well but its readability could be improved with a better presentation of the notations and the model structure. The experimental section is detailed and covers various single-task and multi-task learning applications.
They have compared their solution with other explainable ML algorithms and shown comparable prediction accuracy. My only concern is that their analysis results on interpretability is very heuristic and subjective, rather than a formal validation.

Overall, the paper is a good contribution and I recommend acceptance.

Update (after author response): The authors have addressed my main concerns. I recommend acceptance with the changes proposed in the author response.

**Time Spent Reviewing:**

4

---

> ### Author Response · Authors · 2021-08-09
> **Author response to reviewer oFHP**
>
> We thank the reviewer for their valuable feedback and address their concerns below.
>
> > **My only concern is that their analysis results on interpretability are very heuristic and subjective, rather than a formal validation.**
>
> While interpretability of NAMs may seem heuristic, the graphs of functions learned by NAMs are not just an explanation but an exact description of how NAMs compute a prediction. This is an advantage of NAMs: a decision-maker can easily interpret such models and understand exactly how they make decisions. We illustrated this benefit by validating the behaviour of NAMs on  the MIMIC-dataset (appendix A.1) with a doctor.
>
> Additionally, we argue that inherently interpretable glass-box models like NAMs are superior to black-box models of similar accuracy because global explanations generated for black-box models are approximate and thus not always faithful to how the black-box models computes predictions and do not provide enough detail to fully understand the model’s behavior.
>
> > **The authors should also address some limitations of their current work and potential future avenues of study.**
>
> We’ll add avenues for future research and limitations including possibilities of better activation functions, extensions to domains where GAMs haven’t been applied (e.g, survival analysis [2], ) and incorporating feature interactions.
>
> [1] Rudin, C. (2019). Stop explaining black box machine learning models for high stakes decisions and use interpretable models instead. Nature Machine Intelligence, 1(5), 206-215.
>
> [2] Utkin, L. V., Satyukov, E. D., & Konstantinov, A. V. (2021). SurvNAM: The machine learning survival model explanation. arXiv preprint arXiv:2104.08903.

---

### Decision · Program_Chairs · 2021-09-27

**Decision:**

Accept (Spotlight)

**Comment:**

The reviewers were overall very enthusiastic about this paper, highlighting its novelty, and clarity in terms of writing and visualizations. There are a few changes that I expect you will make based on your responses to the reviewers.